# Dual Function of a Novel Bacterium, *Slackia* sp. D-G6: Detoxifying Deoxynivalenol and Producing the Natural Estrogen Analogue, Equol

**DOI:** 10.3390/toxins12020085

**Published:** 2020-01-26

**Authors:** Xiaojuan Gao, Peiqiang Mu, Xunhua Zhu, Xiaoxuan Chen, Shulin Tang, Yuting Wu, Xiang Miao, Xiaohan Wang, Jikai Wen, Yiqun Deng

**Affiliations:** 1Guangdong Provincial Key Laboratory of Protein Function and Regulation in Agricultural Organisms, College of Life Sciences, South China Agricultural University, Guangzhou 510642, China; gaoxiaojuan@stu.scau.edu.cn (X.G.); mpeiqiang@scau.edu.cn (P.M.); xuanzi@scau.edu.cn (X.C.); shulintang@scau.edu.cn (S.T.); 20182003020@stu.scau.edu.cn (Y.W.); xiangmiao@stu.scau.edu.cn (X.M.); xhwang@stu.scau.edu.cn (X.W.); jkwen@scau.edu.cn (J.W.); 2Key Laboratory of Zoonosis of Ministry of Agriculture, South China Agricultural University, Guangzhou 510642, China; 3Guangdong Laboratory for Lingnan Modern Agriculture, South China Agricultural University, Guangzhou, Guangdong 510642, China

**Keywords:** *Slackia* sp. D-G6, DON, de-epoxidize, equol

## Abstract

Deoxynivalenol (DON) is a highly abundant mycotoxin that exerts many adverse effects on humans and animals. Much effort has been made to control DON in the past, and bio-transformation has emerged as the most promising method. However, useful and effective application of bacterial bio-transformation for the purpose of inhibiting DON remains urgently needed. The current study isolated a novel DON detoxifying bacterium, *Slackia* sp. D-G6 (D-G6), from chicken intestines. D-G6 is a Gram-positive, non-sporulating bacterium, which ranges in size from 0.2–0.4 μm × 0.6–1.0 μm. D-G6 de-epoxidizes DON into a non-toxic form called DOM-1. Optimum conditions required for degradation of DON are 37–47 °C and a pH of 6–10 in WCA medium containing 50% chicken intestinal extract. Besides DON detoxification, D-G6 also produces equol (EQL) from daidzein (DZN), which shows high estrogenic activity, and prevents estrogen-dependent and age-related diseases effectively. Furthermore, the genome of D-G6 was sequenced and characterized. Thirteen genes that show potential for DON de-epoxidation were identified via comparative genomics. In conclusion, a novel bacterium that exhibits the dual function of detoxifying DON and producing the beneficial natural estrogen analogue, EQL, was identified.

## 1. Introduction

Deoxynivalenol (DON) is a mycotoxin that is mainly produced by multiple Fusariums. It contaminates wheat, maize, and other grain crops [1]. DON contamination, which occurs naturally and spontaneously, is highly dependent on climate [2]. Annually, DON causes economic losses in the range of billions of dollars worldwide, by contaminating crops such as wheat and barley [3]. 

DON exerts multiple toxic effects on human and animals. Acute exposure of animals to DON induces abdominal pain, dizziness, headache, throat irritation, nausea, vomiting, diarrhea, and bloody stools [4]. Chronic exposure of animals to DON causes weight loss, anorexia, and nutritional deficiency in animals [5]. The mammalian intestinal tract is a major target of DON. Reportedly, DON impaired the expression of porcine intestinal host defense peptides in weaned piglets, thereby increasing intestinal permeability and significantly decreasing average daily weight gain [6,7]. Additionally, DON targets the innate immune system [8,9]. At the molecular level, many studies have proved that DON may inhibit protein synthesis by interacting with peptidyl transferase, which causes early termination of translation [10,11]. Moreover, DON may also activate mitogen activated protein kinases (MAPKs) [12,13,14], thereby triggering endoplasmic reticulum stress and calcium-mediated signaling [15,16], and/or induce mitochondrial hyperplasia and apoptosis [17,18]. For humans, DON has been classified as a category 3 human carcinogen by the International Agency for Research on Cancer (IARC) [19]. Considering its hazardousness to both humans and animals, the Food and Drug Administration (FDA) has introduced strict limits to levels of DON allowed in foods: 1 ppm for wheat products meant for human consumption, 5–6 ppm on grains or grain byproducts meant for pigs and 10–20 ppm of diets meant for cattle and chicken [3]. Therefore, effective and useful strategies for removing or detoxifying DON are urgently needed, mainly due to the difficulties associated with avoiding contamination by fungi.

Many strategies have been considered for controlling contamination by DON. These include the use of antagonists against DON producing Fusarium spp., [20], cultivation of transgenic crops [21,22,23], preventing damage to grains during harvest, and maintaining critical factors important for storage. The use of physicochemical detoxifying methods that can be applied post contamination, such as baking, radiating, adsorbing, and acid/alkali treatment have also been considered. However, the effects of fungi contamination control or physicochemical detoxification are very limited. Currently, researchers are increasingly focusing on DON bio-transformation, due to its high efficiency, specificity, and environmental-friendliness. For example, a mixed culture D107 can convert DON into 3-keto-4-deoxynivalenol [24]; the soil bacteria *Nocardioides* sp. WSN05-2 [25], *Sphingomonas* S3-4 [26] and *Devosia* mutans 17-2-E-8 [27] degrade DON into 3-epimer-DON (3-epi-DON); *Sphingomonas* KSM1 [28] from lake water catabolizes DON into 16-hydroxy-DON (16-HDON). Both 3-keto-4-deoxynivalenol and 3-epi-DON exhibit little toxicity and 16-HDON is 10-fold less phytotoxic to wheat than DON. Importantly, it has been demonstrated that disrupting the C12-13 epoxide, which plays a key role in DON toxicity, was critical for reducing its toxicity [29,30]. Several bacteria capable of DON de-epoxidation, such as Bacillus sp.LS100 from chicken digesta [31], Biomin BBSH797 (also called DSM 11798) from rumen fluid [32], and *Eggerthella* sp. DII-9 (now revised as *Raoultibacter* sp. DII-9 based on the latest species classification) from chicken intestine [33] have been identified. However, no genes responsible for de-epoxidation of DON have been identified so far. We suppose that it would be a greater chance to clarify the molecular mechanisms involved based on comparative genomics. Therefore, novel bacteria capable of DON de-epoxidation are required in order to clarify the molecular mechanisms involved and to develop effective DON detoxification methods.

The current study screened and identified a novel bacterium with novel function from chicken intestines that detoxifies DON and transforms daidzein (DZN). Metabolic and genomic characteristics of this strain were investigated and potential genes involved in DON de-epoxidation were analyzed.

## 2. Results

### 2.1. Isolating the Bacterium Capable of DON De-Epoxidation from Chicken Intestines

To isolate the bacterium that converts DON to DOM-1, a modified screening procedure (Figure 1a) was set up based on a previous method [28]. After screening and four passages of the subculture, one single colony, *Slackia* sp. D-G6 (D-G6), which showed high activity for DON transformation, was obtained. HPLC analysis showed that D-G6 completely transformed 25 μg/mL DON to DOM-1 after 48 h of co-culture (Figure 1b,c). 

### 2.2. Identification of the Newly Isolated Bacterium, Slackia sp. D-G6 

To identify the isolate, D-G6, 16S rDNA was sequenced. Nucleotide BLAST using a 16S rDNA sequence, showed that the species phylogenetically closest to D-G6, with a sequence similarity of 99.15%, was *Slackia equolifaciens* strain DZE^T^ (accession no. EU377663). Phylogenetic tree analysis further confirmed that D-G6 was placed in the same cluster that included *Slackia* species (Figure 2a). We therefore classified DG-6 as a new species of genus *Slackia*, and named it *Slackia* sp. D-G6 (D-G6).

The morphology of DG-6 was observed via light microscopy and transmission electron microscopy (TEM). Gram staining indicated that D-G6 was a Gram-positive bacterium (Figure 2b). TEM demonstrated that D-G6 varied in size from 0.2–0.4 μm × 0.6–1.0 μm, and was non-sporulating in shape (Figure 2c).

### 2.3. The Metabolic Characteristics of D-G6

To investigate the metabolic characteristics of D-G6 that may be involved in transforming DON, the effects of the medium, pH, and temperature on transforming efficiency were analyzed. Firstly, effects of the medium were analyzed. Transforming efficiency was generally low in all tested media, including NB, BHI, GAM, WCA, and TSB, wherein the highest efficiency (approximately 20%) was observed in WCA (Figure 3a). Interestingly, adding chicken intestine extract (Ext) greatly improved transforming efficiency in all tested media, except GAM (Figure 3b). Especially in WCA medium plus Ext, 25 μg/mL DON was completely transformed to DOM-1 within 48 h (Figure 3b). Thereby, WCA medium plus Ext was chosen as the optimum medium for D-G6 mediated transformation of DON. Secondly, the effects of temperature were analyzed. D-G6 was active at 32–47 °C and inactive at 27 °C and 52 °C (Figure 3c). Moreover, 25 μg/mL of DON was completely transformed into DOM-1 in 24 h at 37 °C, and 42 °C (Figure 3c). Thirdly, the effect of pH was analyzed. D-G6 was active in pH values ranging from 5–11, but showed no activity at pH 3 and 4, indicating that neutral and basic environments were beneficial for DON de-epoxidation (Figure 3d). Finally, the growth and metabolic curves vs. time in WCA medium plus Ext, in pH 6.5, were analyzed. With an inoculation density of approximately 100 CFU/mL, logarithmic growth occurred 24 h after incubation and reached stationary phase 36 h later (Figure 3e). However, DON de-epoxidation did not initiate until the growth reached the stationary phase, and DON was completely transformed 3 days after inoculation (Figure 3e). Next, metabolic activity of D-G6 was compared to that of *Raoultibacter* sp. DII-9 (DII-9) [33]. Under low DON concentrations, the efficacies of D-G6 and DII-9 were comparable, but DII-9 was slightly more efficient than D-G6 when DON concentration was increased (Table 1). 

### 2.4. Slackia sp. D-G6 is Capable of Producing the Natural Estrogen Analogue, EQL, from DZN

As most strains of genus *Slackia* possess the ability to transform DZN to dihydrodaidzein (DHD), tetrahydrodaidzein (THD), or EQL which are beneficial for animal and human health, especially females [34,35,36,37], we investigated whether D-G6 was capable of transforming DZN. Three enzymes, daidzein reductase (DZNR), dihydrodaidzein reductase (DHDR), and tetrahydrodaidzein reductase (THDR), involved in DZN transformation were searched against the genome of D-G6, and all three were detected. DZNR, DHDR, and THDR shared an identity of 100%, 100%, and 99.4%, respectively, with that of *Slackia equolifaciens* strain DZE^T^, which is reported to be a very effective DZN transforming strain [38] 

To verify the capability of D-G6 for transforming DZN, D-G6 was incubated with DZN at 37 °C under anaerobic conditions and the products were detected using HPLC and Q-TOF-MS. Both DZE^T^ and D-G6 converted DZN (5.2 min) to EQL (6.2 min) and DHD (4.2 min) following 2 d of incubation (Figure 4a–d). The 4.2 min product was verified using Q-TOF-MS (Figure 4e,f). When incubation time was prolonged to 3 d, both D-G6 and DZE^T^ completely transformed DZN into EQL (Figure 4g,h). 

The capacity of D-G6 and DZE^T^ was further compared at increasing DZN concentrations. Both D-G6 and DZE^T^ transformed over 90% of 0.6 M DZN to EQL after 3 d of incubation (Table 2). When the concentration was increased to 2 M, both D-G6 and DZE^T^ transformed approximately 50% of 2 M DZN to EQL after 3 d of incubation (Table 2). Altogether, the efficiency displayed by D-G6 for DZN transforming, was comparable to that of the well-known DZN transforming strain, DZE^T^.

### 2.5. Characteristics of the D-G6 Genome and Genes Potentially Involved in DON De-Epoxidation

In order to understand molecular mechanisms underlying DON de-epoxidation, the genome of D-G6 was sequenced and compared to reference genomes of DZE^T^. The genome of DG-6 is 2,863,556-bp long (Figure 5a), with G + C content of 60.48%, 2312 predicted protein coding genes, 29 rRNA genes, and 47 tRNA genes (Table 3). 

To predict the genes potentially involved in DON de-epoxidation, a genome comparison was made between two bacteria capable of DON de-epoxidation (D-G6 and DII-9) and closely related strains which showed no DON de-epoxidation activity DZE^T^ and *Raoultibacter timonensis* P3277 (P3277), respectively. Thirteen clusters were identified as homologous genes between D-G6 and DII-9 that may be involved in DON de-epoxidation (purple color, Figure 5b). The functions of these 13 clusters were predicted and listed (Table 4). Among the 13 genes, two clusters (cluster 892 and 2459) with predicted redox activity and three clusters (cluster 4328, 4382, and 6747) with predicted unknown function were recombinantly expressed and analyzed for DON de-epoxidation activity in vitro. However, all five chosen clusters did not display DON de-epoxidation activity in vitro (data now shown), suggesting that de-epoxidation is a complex process, which may involve multiple enzymes and require special reaction conditions.

## 3. Discussion

Widespread, heavy contamination by DON, has posed a serious threat to the animal feed industry as well as the food industry for a long time. Currently, bio-transformation is considered the most promising strategy for detoxifying DON. The current study identified a novel bacterium, D-G6, capable of DON de-epoxidation. In addition to its role in DON de-epoxidation, D-G6 actively produces a natural estrogen analogue, EQL.

De-epoxidation is one of the most important processes associated with DON detoxification. Previously, we isolated a bacterium, DII-9, which was effective for DON de-epoxidation [33]. Efforts to screen enzymes responsible for DON de-epoxidation in DII-9 have ended in failure. Therefore, we attempted to screen more bacteria, which are capable of DON de-epoxidation, and screened associated enzymes using comparative genomics. As a result, one novel bacterium, D-G6 was identified. Except for DG-6 and DII-9, 3 other bacteria, BBSH797 from bovine rumen [32], LS100 from chicken intestine [31], and *Clostridium* sp. WJ06 (WJ06) from animal intestine [39] have been identified as capable of DON de-epoxidation. BBSH797 belongs to the family *Coriobacteriaceae*. DII-9 and D-G6 belong to the family *Eggerthellaceae*. Interestingly, *Coriobacteriaceae* and *Eggerthellaceae* belong to the same order, Coriobacteriales, suggesting a connection with DON de-epoxidation. However, genome data of BBSH797, LS100, and WJ06 have not yet been revealed, causing difficulties in identifying the enzymes possibly involved in DON de-epoxidation via comparative genomics.

The genome data of four strains were compared. Of these, DG-6 and DII-9, display DON de-epoxidation activity. The other two, DZE^T^, and P3277, which are the strains closest to DG-6 and DII-9, respectively, do not display DON de-epoxidation activity. Thirteen clusters, which are likely involved in DON de-epoxidation were identified (Table 4), five of which were recombinantly expressed and tested. However, all five chosen clusters did not display DON de-epoxidation in vitro (data now shown), suggesting that de-epoxidation is a complex process. The cds-1661 and cds-605 from D-G6 and DII-9, respectively, were annotated as FAD-binding dehydrogenases, and upstream and downstream genes of cds-1695 and cds-795 were FAD oxidoreductases (data not shown, concluded from gene annotations), suggesting that FAD may play a role in DON de-epoxidation. Interestingly, de-epoxidation efficiency increased over 50% when 1 mM FAD, NADPH, or NADH was added to the reaction system comprised of D-G6 membrane components (data not shown), suggesting that the de-epoxidation may be a redox process involving FAD, NADPH, and/or NADH. 

Reportedly, strains of genus *Slackia* have transformed isoflavones effectively. Metabolites of isoflavones are beneficial for human and animal health, and generally function more effectively than their precursors [40,41,42]. Isoflavones mainly contain DZN and genistein, which are present in soybeans and other *Fabaceae*. Because of structural similarities between isoflavones and endogenous estrogens, isoflavones and their metabolites are able to bind estrogen receptors and exert hormonal effects, including anti-oxidative, anti-inflammatory [43], and anti-tumor cell proliferation [44]. When DZN is incubated with D-G6, DZN is effectively degraded into DHD and EQL within 2 days. If incubation occurs for 3 days, complete degradation into EQL is observed (Figure 4 and Table 2). These results indicated that D-G6 activity is comparable to the well-known isoflavone degradation strain DZE^T^ [38]. Clinically, approximately 30% to 50% of humans have the ability to produce EQL from ingested DZN via intestinal bacteria [45].

## 4. Conclusions

In summation, the current study has revealed, for the first time, a dual function bacterium, *Slackia* sp. D-G6, that is active in DON detoxification and EQL production, providing a good resource, which can be used to enhance feed and food additives. D-G6 was a Gram-positive non-sporulating bacterium varying from 0.2–0.4 μm × 0.6–1.0 μm. The optimum conditions for D-G6 to transform DON were within 37–47 °C and pH 6–10.

## 5. Materials and Methods 

### 5.1. Materials

DON and DOM-1 were purchased from Pribolab (Beijing, China) and Sigma-Aldrich (St. Louis, MO, USA), respectively. Daidzein and equol were purchased from ADOOQ (Irvine, CA, USA) and SellckChem (Houston, TX, USA), respectively. Acetonitrile (HPLC grade) was obtained from CNW Technologies (Duesseldorf, Germany). Brain heart infusion was purchased from Thermo Fisher Oxoid (Altrincham, England). NB, GAM, WCA, TSB, and Columbia were all obtained from Qingdao haibo biology company (Qingdao, China).

### 5.2. Bacterium Screening Methods 

One gram of feces from chicken intestines was mixed with 1 ml BHI broth to produce a homogeneous mixture. The mixture (20 μL) was inoculated with 180 μL fresh of BHI and 25 μg/mL of DON, and incubated at 37 °C under anaerobic conditions. The total concentration of DON used in this study was 25 μg/mL unless otherwise specified. DON and its product DOM-1 were detected 3 d following incubation using HPLC. Cultures with transforming rates over 50% were selected as samples for subsequent screening. Selected cultures were freshly inoculated with dilution of 10–10^6^ fold in series. Subsequently, cultures with transformation rates >50% and inoculation densities lower than 10^3^ CFU/ml were selected as samples for the next screening iteration. Single colonies were isolated from BHI agar and their activity was confirmed. Culture media were then screened to improve the activity of colonies that showed DON transforming ability. NB, GAM, WCA, TSB, and Columbia with or without chicken intestinal extract (Ext) were analyzed.

### 5.3. Extraction and Analysis of DON and DOM-1

Detection of DON and DOM-1 has been described previously [33]. In brief, after 3 d of incubation, 3 times the volume of ethyl acetate was added to the culture and the resultant mixture was subjected to vibration for 1 min. The supernatant was evaporated using a stream of highly pure nitrogen, until dry. Next, 25% methanol (200 μL) was added to dissolve the dried samples. Extracted samples were detected via HPLC using previously described [33].

### 5.4. Comparing the Transforming Rate under Different Culture Media, Temperatures and pH

Five basic media (BHI, NB, GAM, WCA, and TSB) and their modified forms (addition of 50% Ext) were analyzed. Aliquots (2 μL) of the *Slackia* sp. D-G6 (D-G6) culture sample were transferred to 0.2 ml of medium, and incubated under anaerobic conditions for 3 days. Following incubation, DON and DOM-1 were extracted and analyzed using HPLC. The transforming rate was calculated as DOM-1/DON+DOM-1 in all experiments.

Six temperature, 27, 32, 37, 42, 47, and 52 °C, and 9 pH conditions, 3, 4, 5, 6, 7, 8, 9, 10, and 11 were analyzed. D-G6 was inoculated into 0.2 ml WCA medium plus 50% Ext at an initial density of 10^7^–10^8^ CFU/mL. The samples were incubated under anaerobic conditions for 3 days. Following incubation, the DON and DOM-1 were extracted and analyzed. Curves of growth and metabolic activity vs. time were analyzed in WCA medium plus 50% Ext, under 37 °C and pH 6.5.

### 5.5. Phylogenetic Tree Construction

The 16S rDNA of D-G6 was amplified using primers 27f and 1492r [38,46]. BLAST analysis available at the National Center for Biotechnology Information (NCBI) database was used to analyze the sequence similarity (https://blast.ncbi.nlm.nih.gov/Blast.cgi?PROGRAM=blastn&PAGE_TYPE=BlastSearch&LINK_LOC=blasthome). Sequences that were either phylogenetically closest or functionally similar were downloaded. The phylogenetic tree was constructed using MEGA 6 software [47]. Bootstrap values were calculated based on 1000 re-samplings via the neighbor-joining method.

### 5.6. Morphologic Analysis

Morphology was analyzed via gram staining and Transmission Electron Microscopy (TEM). Following gram staining, an optical microscope with 100× oil immersion was used to observe bacterial cells revived in BHI agar for 1–2 days. Bacteria in the logarithmic phase were washed with PBS and fixed using phosphomolybdic acid. Subsequently, the size and morphology of D-G6 were observed using TEM (Tecnai 12, FEI company, Eindhoven, Netherlands).

### 5.7. Genome Sequencing

The genome of D-G6 was sequenced by GENEWIZ incorporated company (Shenzhen, China). One hundred nanograms of D-G6 genomic DNA was randomly fragmented to <500 bp via sonication (Covaris S220). The fragments were modified via 5′ phosphorylation and dA-tailing. Next, adaptors were added to both ends after the fragments. Fragments of ~470 bp were purified by size based selection of adaptor-ligated DNA. The library was amplified via PCR for 8 cycles using primers P5 and P7. Following clean up and validation using an Agilent 2100 Bioanalyzer (Agilent Technologies, Palo Alto, CA, USA) and quantified by Qubit 3.0 Fluorometer (Invitrogen, Carlsbad, CA, USA), sequencing was carried out using a 2 × 150 paired-end (PE) configuration and monitored by the HiSeq Control Software (HCS) + OLB + GAPipeline-1.6 (Illumina) (San Diego, CA, USA) on a HiSeq instrument and processed via image analysis and base calling.

For analysis performed by Pacbio (Menlo Park, CA, USA), the construction of the SMRTbell library required 10 kb double-stranded DNA fragments end repaired and ligated with universal hairpin adapters. The library was sequenced in a PacBio RSII/Sequel SMRT instrument [48]. PacBio reads were assembled using HGAP4/Falcon WGS-Assembler 8.2 [49], and re-corrected through either Pilon software using previous Illumina data or Quiver using Pacbio reads.

### 5.8. Drawing the Genome Circle Map

The genome circle map was drawn by using of Circos (version 0.69, Genome Sciences Centre, Vancouver, BC, Canada) software. A circle diagram, containing 7 rings, of the D-G6 genome. The first circle shows the size of the genome, the second circle represents GC content, the third circle shows coding genes on the positive strand (red), the fourth circle shows coding genes on the negative strand (green), the fifth and sixth circles represent ncRNA on the positive strand (blue) and the negative strand (purple), respectively and the seventh circle presents information on repeating sequences of long segments within the genome (orange).

### 5.9. Comparing the Activity of Slackia sp. D-G6 and Slackia equolifaciens Strain DZE^T^ (DZE^T^) on Transforming Daidzein (DZN) 

D-G6 and DZET were inoculated into the medium (WCA+ 50% intestine extract) treated with 0.2 M DZN, and incubated under anaerobic conditions at 37 °C. Following a 3 days incubation period, ethyl acetate (600 μL) was added to stop the reaction following which the product was vortexed vigorously. The organic phase was separated and transferred to a new Eppendorf tube, vacuum dried, and dissolved in 0.2 ml of methanol. Extracted samples were analyzed via HPLC and UPLC- Q-TOF-MS to quantify levels of DZN and its potential metabolites [50]. Metabolites were analyzed using a C18 reverse column (5 μm, 4.6 × 250 mm) with an isocratic mobile phase comprising, 45% water: acetic acid (98:2, *v*/*v*) in methanol (1 mL/min, 40 °C) with detection at 280 nm [50]. For UPLC-Q-TOF-MS, an Agilent Eclipse plus C18 (1.8 μm, 50 mm × 2.1 mm) was used, and the mobile phase was comprised of 60% water: formic acid (98:0.2, *v*/*v*) in methanol (0.3 mL/min). Gas temperature was 300 °C with 8 L/min gas flow.

## 6. Patents

There is a patent on pending.

## Figures and Tables

**Figure 1 toxins-12-00085-f001:**
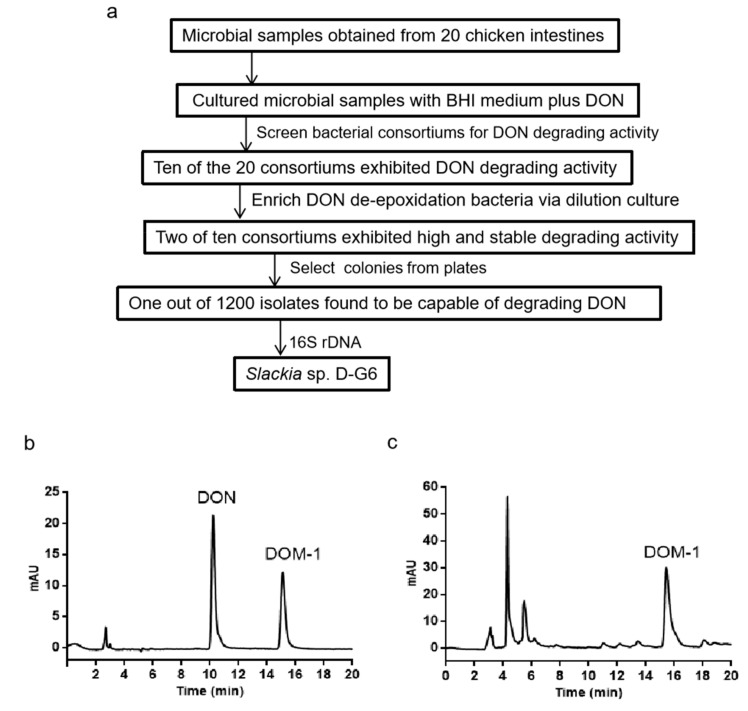
Screening procedure and HPLC analysis of DON and DOM-1 after co-culture of DON with D-G6. (**a**). Procedure for screening DON transforming bacterium. (**b**). HPLC chromatography of standard DON and DOM-1. (**c**). The products of DON transformed by D-G6. DON and DOM-1 standards are indicated at 10.1 min and 15.6 min, respectively.

**Figure 2 toxins-12-00085-f002:**
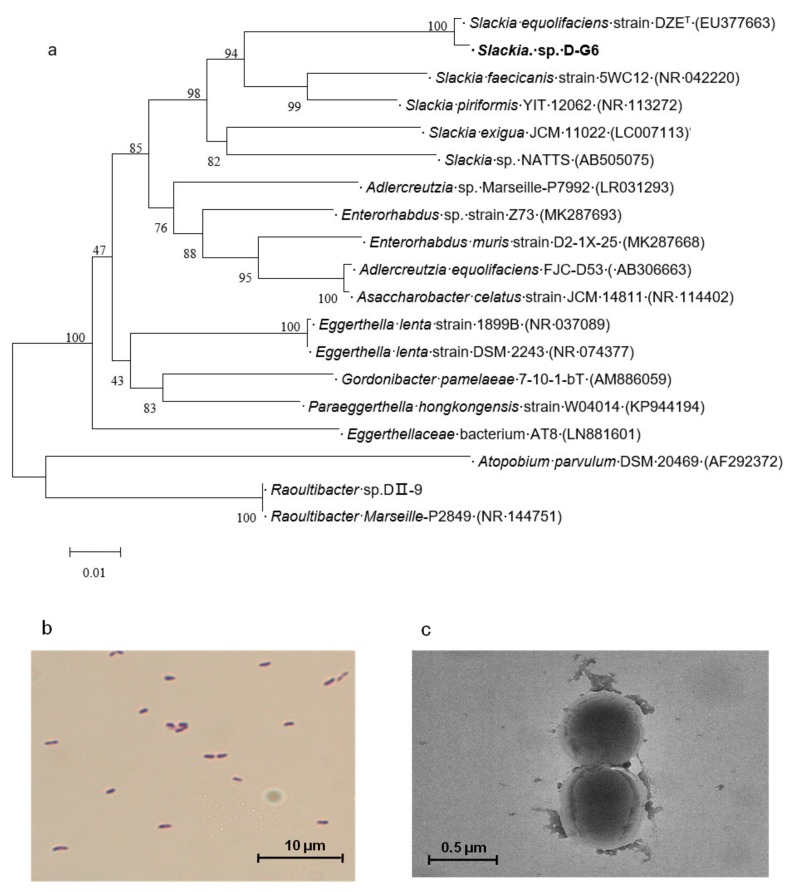
The phylogenetic tree and morphology of *Slackia* sp. D-G6. (**a**). The phylogenetic tree of D-G6 based on 16S rDNA genes. The GenBank accession numbers of the sequences are shown in parentheses. The bar indicates 0.01 substitutions per nucleotide position. (**b**). Gram staining of D-G6. (**c**). Microscopic images of D-G6 under TEM.

**Figure 3 toxins-12-00085-f003:**
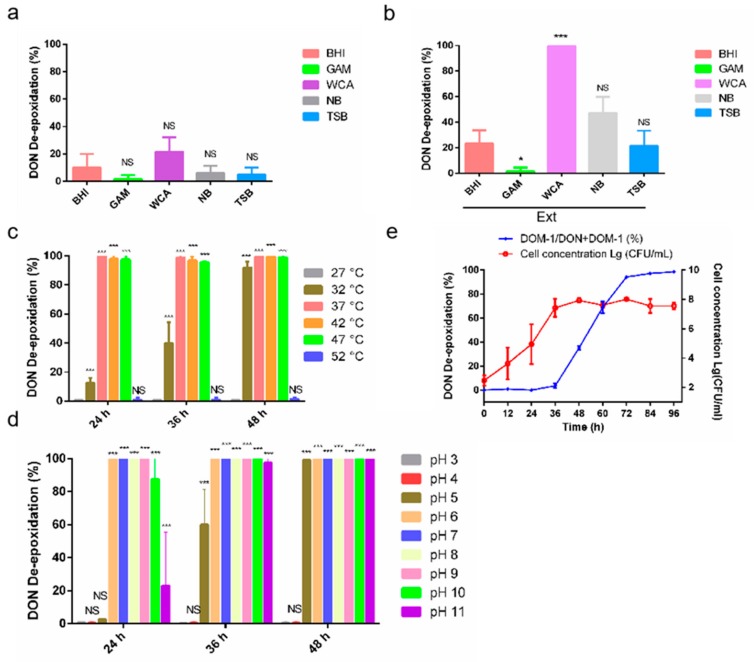
Metabolic characteristics of *Slackia* sp. D-G6. (**a**). Transforming efficiency under different media. (**b**). Transforming efficiency under different medias plus chicken intestine extracts (Ext). (**c**). The efficiency of DON de-epoxidation of D-G6 at different temperatures. The experiments were performed in WCA modified medium at pH 6.5. (**d**). The effects of varying pH on DON de-epoxidation. The experiments were performed in WCA modified medium and the temperature was set at 37 °C. (**e**). The growth curve and metabolic curve of D-G6 against time. Inoculation density was approximately 100 CFU/mL, the temperature was 37 °C, and pH was 6.5. All experiments were replicated three times biologically. The error bars represent the standard deviations. * indicate significantl difference (* *P* < 0.05, *** *P* < 0.001), “NS” means “no significance”.

**Figure 4 toxins-12-00085-f004:**
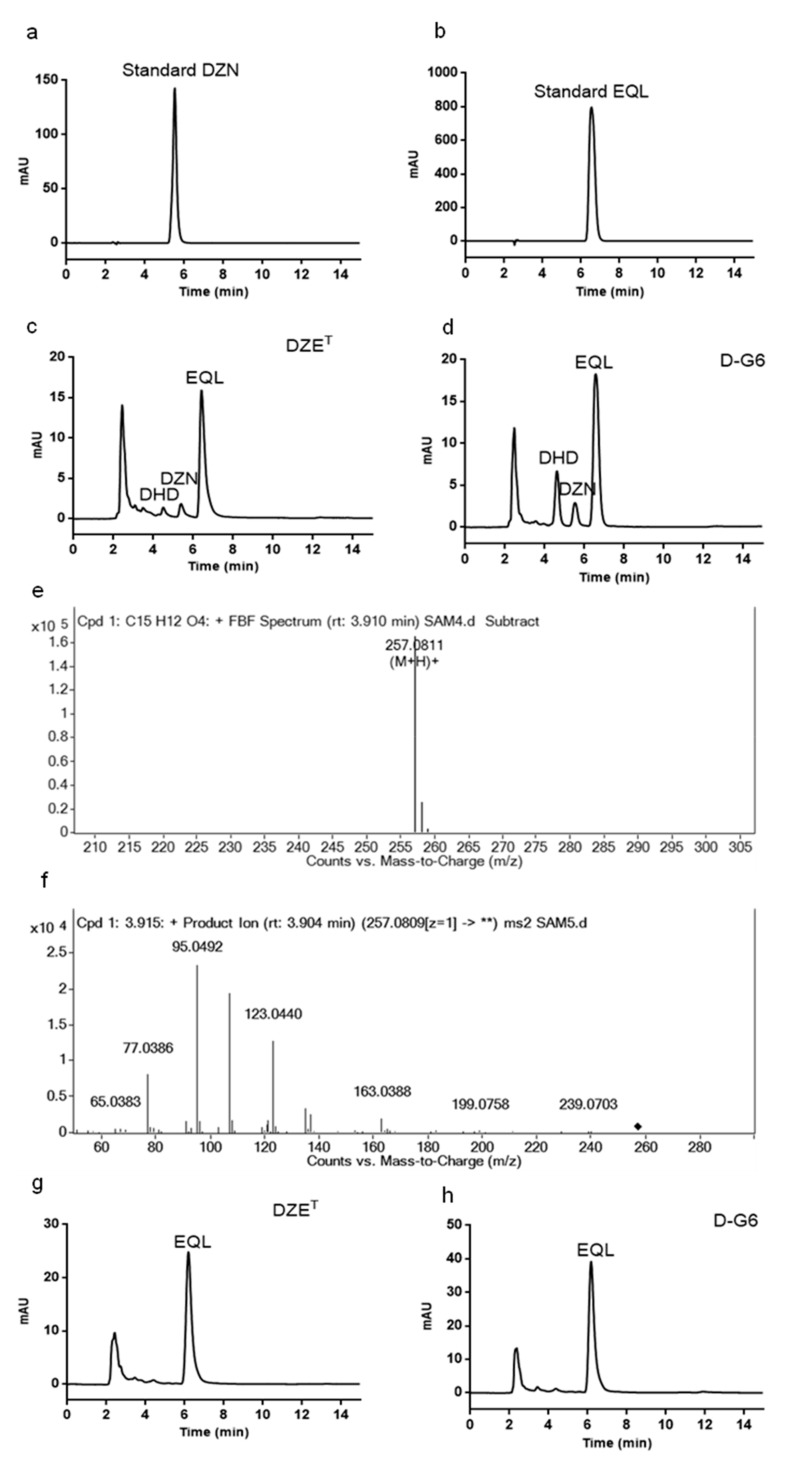
*Slackia* sp. D-G6 is capable of transforming daidzein (DZN). (**a**,**b**). DZN standards (**a**) and equol (EQL) (**b**). The retention time of DZN and EQL were approximately 5.2 min and 6.2 min, respectively. (**c**,**d**). HPLC analysis of the products after 2 d of incubation of DZE^T^ (**c**) or D-G6 (**d**) with DZN. (**e**,**f**). Mass spectrometry (**e**) and mass/mass spectrometry (**f**) analysis of the product at 4.2 min. g,h. HPLC analysis of the products after 3 d of incubation with DZE^T^ (**g**) or D-G6 (**h**) with DZN.

**Figure 5 toxins-12-00085-f005:**
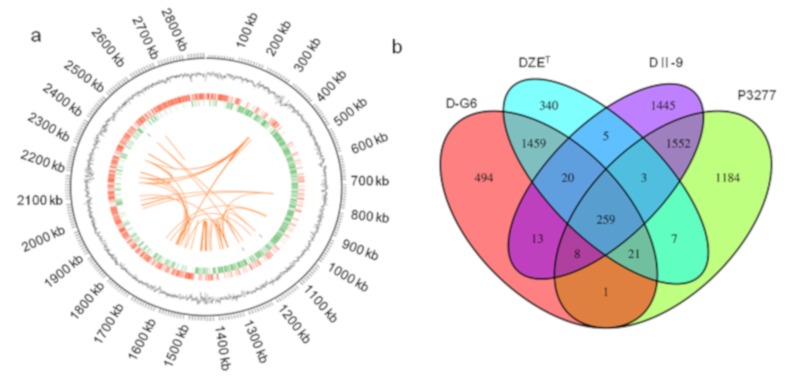
The genome of *Slackia* sp. D-G6 and comparative analysis of four genomes, D-G6, DZE^T^, DII-9, and P3277. (**a**). The genome circle map was drawn using Circos (version 0.69) software. Seven types of information are contained in the circle diagram, from outside to inside: the first circle shows the size of the genome, the second one is GC content, the third one are coding genes on the positive chain (red), the fourth one are the coding genes on the negative chain (green), the fifth and sixth ones are ncRNA on positive strand (blue) and negative strand (purple) respectively, and the seventh one are the repeating sequences of long segments within the genome (orange). (**b**). Venn diagram showing the core genes and genes shared among D-G6, DII-9, DZE^T^, and P3277.

**Table 1 toxins-12-00085-t001:** Comparison of DON de-epoxidation efficiency between D-G6 and DII-9.

DON (μg/mL)	Time (d)	DOM-1/DON+DOM-1 (%)	
		D-G6	DⅡ-9
25	1	100	100
	2	100	100
75	1	60–70	80–100
	2	>95	>95
250	1	20–35	40–50
	2	>90	>95

Note: The concentration of D-G6 or DⅡ-9 were 10^7^–10^8^ CFU/mL in 0.2 mL WCA modified medium. Incubation conditions were as follows: 37 °C and pH 6.5. The experiments were repeated at least 3 times independently.

**Table 2 toxins-12-00085-t002:** Comparison of the efficiencies of D-G6 and DZE^T^ for transforming DZN to EQL.

DZN (M)	Time (d)	D-G6			DZE^T^		
		DHD (%)	DZN (%)	EQL (%)	DHD (%)	DZN (%)	EQL (%)
0.2	2	5–30	0–10	60–100	0–18	0–6	89–100
	3	0–5	0	>97	0–6	0	>90
0.6	2	10–30	30–40	32–52	8–18	10–19	60–76
	3	4–9	<5	>90	0	<5	>95
2	2	3–15	68–80	10–19	2–10	60–78	14–25
	3	8–15	20–35	50–60	5–10	25–40	55–65

Note: 10^7^–10^8^ CFU/mL of D-G6 or DZE^T^ were inoculated into 0.2 mL WCA modified medium. The conditions for incubation were as follows: 37 °C, pH 7.2. All experiments were repeated at least 3 times independently.

**Table 3 toxins-12-00085-t003:** Genome comparison between D-G6 and DZE^T.^

	D-G6	DZE^T^
Size (Mb)	2.86	2.75
G + C (%)	60.48	59.8
Total number of protein-coding genes	2312	2149
Total number of rRNA genes	29	10
Total number of tRNA genes	47	47

**Table 4 toxins-12-00085-t004:** The list of 13 clusters that may be critical for DON de-epoxidation.

Clusters	Predicted Functions	cds Number of D-G6	cds Number of D II-9
Cluster-761	potassium-transporting ATPase activity	cds-765	cds-727
Cluster-892	FAD-binding dehydrogenase	cds-1661	cds-605
Cluster-1703	ISL3 family transposase	cds-1137/1145/1147/1172/1175	cds-1991/1993
Cluster-2351	IS256 family transposase	cds-761/1023/1046	cds-1169/1999
Cluster-2392	ring-opening amidohydrolase	cds-1648	cds-1300
Cluster-2459	catalytic activity | iron-sulfur cluster binding	cds-1695	cds-795
Cluster-3749	aminoglycoside 3′-phosphotransferase III	cds-1146	cds-1992
Cluster-3864	aminoglycoside 3″-adenylyltransferase activity| response to antibiotic	cds-2270	cds-1167
Cluster-4013	dimethyladenosine transferase	cds-1144	cds-1932
Cluster-4328	protein of unknown function	cds-1907	cds-1995
Cluster-4382	hypothetical protein	cds-2273	cds-1837
Cluster-6654	isoleucyl-tRNA synthetase	cds-1205	cds-928
Cluster-6747	protein of unknown function	cds-2271	cds-1166

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
