# Peer review of "Dual Function of a Novel Bacterium, *Slackia* sp. D-G6: Detoxifying Deoxynivalenol and Producing the Natural Estrogen Analogue, Equol"

_toxins, 2020, doi:10.3390/toxins12020085_

Round 1
Reviewer 1 Report
The manuscript presents study on detoxification of deoxynivalenol using bacterium isolated from chicken intestine. The authors described method of isolation of detoxifying bacteria, characterized its activity and described genes potentially involved in DON detoxification. The manuscript is scientifically sound and in my opinion should be published after minor revision.
Comments:
Line 104: 37 degrees appear twice, probably an error;
Line 110: "days" instead of "d" would be easier to understand
Figure 3c and 3d are difficult to read
Table 4 should be rearranged
Conclusions should be expanded
Line 238: "10^6" instead of "106"
The manuscript lacks statistical analysis of data.
The description of genome circle map should be placed in figure caption.
Author Response
Point 1: Line 104: 37 degrees appear twice, probably an error;
Response 1: Sorry for the typo error. We corrected it as indicated in page 4, line 113 in the track change version of revised manuscript.
Point 2: Line 110: "days" instead of "d" would be easier to understand.
Response 2: Thank you for your suggestion. We correct it as indicated in page 5, line 120 in the track change version of revised manuscript and checked throughout the manuscript.
Point 3: Figure 3c and 3d are difficult to read.
Response 3: Thank you for your comments. We revised the figure 3 c and 3d.
Point 4: Table 4 should be rearranged.
Response 4: Thank you for your comments. We revised table 4.
Point 5: Conclusions should be expanded.
Response 5: Thank you for your comments. We revised the conclusions as indicated in page 11, line 241-245 in the track change version of revised manuscript.
Point 6: Line 238: "10^6" instead of "106"
Response 6: Sorry for the typo error. We corrected it as indicated in page 12, line 260 in the track change version of revised manuscript.
Point 7: The manuscript lacks statistical analysis of data.
Response 7: Thank you for your comments. The data of figure 3 were statistically analyzed and revised.
Point 8: The description of genome circle map should be placed in figure caption.
Response 8: Thank you for your comments. We revised it as indicated in page 9, line 187-192 in the track change version of revised manuscript.
Reviewer 2 Report
In the article entitled “Dual function of a novel bacterium, Slackia sp. D-G6: Detoxifying deoxynivalenol and producing the natural estrogen analogue, equol”, authors describe the isolation and identification of a Slackia sp. D-G6 bacteria from chicken intestine. It is Gram positive non sporulating, and has the ability to de-epoxidize DON into a non-toxic form called DOM-1.
Authors also showed that D-G6 also produces equol from Daizein, with high estrogenic activity- prevents estrogen-dependent and age-related diseases effectively.
Finally, the Genome sequence let to the authors to identify- 13 potential de-epoxidation genes.
The article is interesting, and experiments well performed and described. However, this reviewer found some weakness regardless the originality of the study, since other bacteria were previously described with similar activities to those described in this article. Furthermore, the conclusion reached as result of the genome sequences are too speculative. Thus, in order to increase the interest of this article, authors would have to do an effort to solves these two points.
Other comments and suggestions are included below.
Introduction
Clearly explain the importance of DON as food contaminant with important toxic effects on human and animals. Also introduce some of the strategies used until now to bio-transform DON to a non-toxic derivative.
The question at this point is that the approach described in the current work seems to be very similar to that followed by other previous studies. Please, could you include in the Introduction some clues to understand the differences of this studies with those already included in this section. i.e. those included in references 26, 27 and 28.
Results
-In Figure 1, authors indicate that one out of thousands of isolates found to be capable of degrading DON.- Thus, could authors be more accurate indicating an approximate number of isolates analyzed?
- Figure 3 (a-e). Why the Y-axis maximum value is 120%? When the maximum value can not exceed 100%. Please, adapt the graphs to a 100% value in the Y-axis.
- Figure 5b. in this Figure the 13 clusters common between DII-9 and D-G6 strains indicated that are homologous genes, but not that they are involved in DON de-epoxidation. Some biological evidence would be needed to confirm this hypothesis. Even when some of them encoded for predicted functions related to DON-like compounds detoxification, the sequence data are not enough to reach this conclusion.
Author Response
Point 1: Introduction——Clearly explain the importance of DON as food contaminant with important toxic effects on human and animals. Also introduce some of the strategies used until now to bio-transform DON to a non-toxic derivative.
Response 1: Thank you for your suggestions. We revised it as indicated in page 1 and 2 in the track change version of revised manuscript.
Point 2: Introduction——The question at this point is that the approach described in the current work seems to be very similar to that followed by other previous studies. Please, could you include in the Introduction some clues to understand the differences of this studies with those already included in this section. i.e. those included in references 26, 27 and 28.
Response 2: Thank you for your comments. The differences are that the identified bacterium is a novel species with novel function that detoxifies DON and transforms Daidzein. We revised the last paragraph of the introduction.
Point 3: Results——In Figure 1, authors indicate that one out of thousands of isolates found to be capable of degrading DON.- Thus, could authors be more accurate indicating an approximate number of isolates analyzed?
Response 3: Thank you for your comments. 1200 isolates were analyzed. We revised it in figure 1.
Point 4: Results——Figure 3 (a-e). Why the Y-axis maximum value is 120%? When the maximum value can not exceed 100%. Please, adapt the graphs to a 100% value in the Y-axis.
Response 4: Thank you for your suggestion. We revised Figure 3 as you suggested.
Point 5: Results——Figure 5b. in this Figure the 13 clusters common between DII-9 and D-G6 strains indicated that are homologous genes, but not that they are involved in DON de-epoxidation. Some biological evidence would be needed to confirm this hypothesis. Even when some of them encoded for predicted functions related to DON-like compounds detoxification, the sequence data are not enough to reach this conclusion.
Response 5: Thank you for your comments. We recombinantly expressed 5 of these clusters and analyzed their activity in vitro with negative results. We propose that the de-epoxidation of DON may be a complex process, which may involve multiple enzymes and require special reaction conditions. Currently, we haven’t gotten positive result. So, we didn’t include these data in the manuscript. We modified this part as indicated in page 8, line 177 in the track change version of revised manuscript.
Point 6: The article is interesting, and experiments well performed and described. However, this reviewer found some weakness regardless the originality of the study, since other bacteria were previously described with similar activities to those described in this article. Furthermore, the conclusion reached as result of the genome sequences are too speculative. Thus, in order to increase the interest of this article, authors would have to do an effort to solves these two points.
Response 6: Thank you for your comments. We understand your concerns. The originality of this study is that the identified bacterium, D-G6, is a novel species, which possess the activity of DON de-epoxidation, and also transforms daidzein into equol, which is beneficial for animal and human health. The genomic data provides useful information related to the molecular mechanism of DON de-epoxidation, because the identified 13 clusters are only found in the active strains, D-G6 and DII-9, but not in the inactive and phylogenic closest stains, DZET and P3277. However, our preliminary data showed that multiple enzymes and special reaction conditions may be required in the DON de-epoxidation. It still needs long time to clarify the exact molecular mechanism and which may be reported in out next paper.
Round 2
Reviewer 2 Report
Authors have addresses all comments made by this reviewer. Thus, I think it can be accepted in the present form.